# Sensitivity of Nitrate Aerosol Production to Vehicular Emissions in an Urban Street

**Minjoong J. Kim**

Department of Environmental Engineering and Energy, Myongji University, Yongin 17058, Gyunggi, Korea; minjoongkim@mju.ac.kr

**Abstract:** This study investigated the sensitivity of nitrate aerosols to vehicular emissions in urban streets using a coupled computational fluid dynamics (CFD)–chemistry model. Nitrate concentrations were highest at the street surface level following $NH_3$ emissions from vehicles, indicating that ammonium nitrate formation occurs under $NH_3$-limited conditions in street canyons. Sensitivity simulations revealed that the nitrate concentration has no clear relationship with the $NO_x$ emission rate, showing nitrate changes of only 2% across among 16 time differences in $NO_x$ emissions. $NO_x$ emissions show a conflicting effect on nitrate production via decreasing $O_3$ and increasing $NO_2$ concentrations under a volatile organic compound (VOC)-limited regime for $O_3$ production. The sensitivity simulations also show that nitrate aerosol is proportional to vehicular VOC and $NH_3$ emissions in the street canyon. Changes of VOC emissions affect the nitrate aerosol and $HNO_3$ concentrations through changes in the $O_3$ concentration under a VOC-limited regime for $O_3$ production. Nitrate aerosol concentration is influenced by vehicular $NH_3$ emissions, which produce ammonium nitrate effectively under an $NH_3$-limited regime for nitrate production. This research suggests that, when vehicular emissions are dominant in winter, the control of vehicular VOC and $NH_3$ emissions might be a more effective way to degrade $PM_{2.5}$ problems than the control of $NO_x$.

**Keywords:** Urban pollution; Street canyon; Nitrate aerosol; CFD; Air quality

## 1. Introduction

Nitrate aerosol is a fine particulate matter ($PM_{2.5}$) component produced from the reaction of gas-phase nitrate (nitric acid; $HNO_3$) and ammonia ($NH_3$). During a haze event, nitrate aerosol often contributes to the total observed particle mass by as much or more than the organic fraction across East Asia [1–4]. As severe haze events have increased across East Asia [5], nitrate contributions to $PM_{2.5}$ mass have also increased in polluted urban areas [1,6].

The production of nitrate aerosol in urban areas is affected by vehicular emissions such as nitrogen oxides ($NO_x = NO + NO_2$) and $NH_3$ [7–9]. These vehicular emissions are highly concentrated and are transported by turbulence from the complex geometry of buildings, causing steep gradients of pollutant concentrations [10–13]. Nitrate aerosol chemistry is highly nonlinear and follows the concentrations of precursor gases and humidity, so the formation and distribution of nitrate aerosol are not spatially uniform [14,15]. Most modeling studies on nitrate aerosols are based on regional or global air-quality models that have clear limitations due to their coarse resolution [16–18]. Therefore, to accurately investigate nitrate formation in urban areas, fine-scale simulations that can conserve highly concentrated emission plumes and turbulence are necessary.

Meanwhile, policies regarding vehicular nitrate control focus on reduction of $NO_x$ (e.g., through banning diesel vehicles) [19,20], based on studies of regional or global air-quality modeling. However, studies from observation campaigns have often reported that other vehicle emissions are

much more important than $NO_x$ emissions for nitrate production in urban areas [21–23]. Link et al. investigated the secondary formation of ammonium nitrate from vehicle exhaust using sampling and laboratory experiments in the Seoul Metropolitan Region (SMR) [21]. They found that the secondary production of ammonium nitrate from diesel exhausts is much lower than that from gasoline. They concluded that the $NH_3$ source from gasoline vehicles could be more important than $NO_x$ emissions, indicating that SMR is an $NH_3$-limited environment. Wen et al. studied nitrate formation during a severe $PM_{2.5}$ pollution period and reported that high $NH_3$ concentrations in the early mornings significantly accelerated the formation of fine particulate nitrate [23]. In addition, they found that the increased rate of nitrate aerosol had a strong positive correlation with ozone ($O_3$) concentrations at night, indicating the essential role of oxidants in nitrate formation. Studies also reported that volatile organic compound (VOC) concentrations control nitrate formation by affecting the $O_3$ levels [22,24]. These studies indicate that nitrate aerosols are affected by a complex chemical condition that involves $NO_x$, $NH_3$, and oxidants, whereas policy tends to focus exclusively on $NO_x$ control. Thus, understanding the favorable conditions for nitrate formation in urban areas is crucial for the design of air-quality policies.

Thus, this study investigates the distribution of nitrate aerosols and the favorable conditions for nitration formation in urban streets using a microscale coupled computational fluid dynamics (CFD)–chemistry model that can reproduce the turbulence from complex building geometries. Sensitivity simulations were conducted to examine the sensitivity of emissions to nitrate production by changing the emissions of precursor gases for nitrate aerosols and oxidants. The sensitivity simulation results reveal what significant factors lead to nitrate aerosol problems in urban streets.

## 2. Model Description and Simulation Set-Up

### 2.1. Model Description

A coupled CFD–chemistry model was used based on that proposed by Kim et al. [25]. The CFD model is based on the Reynolds-averaged Navier–Stokes equation (RANS) model and assumes a three-dimensional (3-D), nonrotating, nonhydrostatic, and incompressible airflow system [26]. This model was previously used to examine the flow and dispersion of both reactive gas pollutants [25,27] and reactive aerosol pollutants [25,28].

The model's chemical mechanism includes a full tropospheric $NO_x$–$O_x$–VOC chemistry scheme from a global 3-D chemical transport model (GEOS-Chem V11-1) [29]. GEOS-Chem was initially developed to solve global air chemistry issues; however, application of the GEOS-Chem model has now been extended to the regional scale. The GEOS-Chem model can successfully explain urban air quality, including cases of severe haze over East Asia [1,30,31]. The chemical scheme includes 140 species and 393 reactions, among which 61 reactions are photochemical. Among the 140 species simulated in the chemistry module, the CFD model transports 65 chemical tracers. Radical species with very short chemical lifetimes are not transported. Photolysis rate coefficients are calculated using the Fast-JX radiative transfer model [32,33].

The model also calculates aerosols that include sulfate, nitrate, ammonium, black carbon, and organic carbon [34,35]. Sulfate formation generally occurs via two pathways: the gas-phase oxidation of $SO_2$ by OH and the aqueous-phase oxidation of $SO_2$ by ozone and hydrogen peroxide. The CFD model only accounts for the gas-phase oxidation of $SO_2$ by OH because it lacks an atmospheric physics module that simulates hydrometeors such as clouds and rain.

Nitrate and ammonium aerosol were calculated by partitioning the total $NH_3$ and $HNO_3$ between the gas and aerosol phases. We used the ISORROPIA-II model as a thermodynamic equilibrium model for aerosol partitioning [36,37] and employed it to calculate the thermodynamic equilibrium of a $K^+$–$Ca^{2+}$–$Mg^{2+}$–$NH_4^+$–$Na^+$–$SO_4^{2-}$–$NO_3^-$–$Cl^-$–$H_2O$ aerosol system based on the $NH_3$, $HNO_3$, and $SO_4^{2-}$ concentrations.

The model also includes the production of $HNO_3$ via heterogeneous chemistry between aerosols and gases following Jacob (2000) [38]. The reactions that contribute to $HNO_3$ production by heterogeneous chemistry can be written as:

$$2\,NO_2 \rightarrow HNO_3 + HONO \tag{R1}$$

$$NO_3 \rightarrow HNO_3 \tag{R2}$$

$$N_2O_5 \rightarrow 2\,HNO_3. \tag{R3}$$

The uptake coefficients, $\gamma$, of R2 and R3 were $10^{-4}$ and 0.1, respectively, following Jacob (2000) [38]. For R4, $\gamma$ was set to 0.01 as suggested by Zhang et al. (2012) and Walker et al. (2012) [39,40]. $N_2O_5$ is essential for night-time nitrate aerosol chemistry; the production and loss reactions of $N_2O_5$ can be written as:

$$NO_2 + NO_3 + M \rightarrow N_2O_5 + M \tag{R4}$$

$$N_2O_5 + M \rightarrow NO_2 + NO_3 + M \tag{R5}$$

$$N_2O_5 + h\nu \rightarrow NO_3 + NO_2. \tag{R6}$$

The simulation of carbonaceous aerosols follows the GEOS-Chem model [35]. The primary carbonaceous aerosol follows the passive tracer without any chemical reactions. However, the model resolves primary Black Carbon (BC) and Organic carbon (OC) with a hydrophobic and a hydrophilic fraction for each (i.e., making four aerosol types) for deposition processes. All sources emit hydrophobic aerosols that then become hydrophilic with an e-folding time of 1.2 days, as per Cooke et al. (1999) [41]. Although secondary organic aerosol (SOA) chemistry is not considered in the model, we treat boundary inflow and the transport of SOAs. However, the model does not account for either dust or sea-salt aerosol.

The dry deposition of gases and aerosols was simulated with a standard big-leaf resistance-in-series model [42]. The model accounts for the dry deposition of 46 species, including aerosols. The CFD model has no atmospheric physics module that simulates hydrometeors (e.g., clouds and rain); thus, wet deposition was not calculated, following Kim et al. (2012) [25].

## 2.2. Simulation Set-Up

We assumed a street canyon selected for a simulation located in SMR, one of the most polluted cities, to investigate nitrate formation in urban streets. The domain size was $120 \times 80 \times 100$ m in the x, y, and z directions, respectively. The grid intervals in all directions were 2 m, and the building height was 20 m with unified aspect ratios for all street canyons. Figure 1 shows the detailed structure of the simulation domain.

First, we set a control run (CNTL hereafter) with standard emissions. We estimated the pollutant emissions from average traffic volume in 2017 obtained from the Traffic Monitoring System (http://www.road.re.kr), which provides traffic statistics in SMR. On that basis, the daily traffic volume for each street was assumed to be 15,130 vehicles $day^{-1}$. The monthly averaged diurnal variation in traffic volume was also obtained from the Traffic Monitoring System. Vehicular emissions were computed using the emission factor from the Clean Air Policy Support System (CAPPS) emission inventory [43] and calculated by multiplying the mean ratios of vehicle sizes in Korea. The emission factors following vehicle size and the ratios of vehicle size are summarized in Tables 1 and 2, respectively. The calculated averaged emissions per vehicle were 0.10 g $km^{-1}$, 0.15 g $km^{-1}$, 0.015 g $km^{-1}$, and 0.011 g $km^{-1}$ for $NO_x$, CO, VOC, and $NH_3$, respectively. $NO_x$ emissions were separated into NO and $NO_2$ emissions at a 10:1 ratio by volume [44]. Total VOC emissions were further speciated using the method developed by EMEP/EEA (2016) [45]. Table 3 lists the emission rate and ratios of speciated VOC in the CNTL simulation. All vehicular emissions were emitted at z = 1 m from 4 m wide area sources located at the center of the streets. Other emissions in the model domain were not considered

in the simulation. Previous studies have reported that the largest proportion of emissions in SMR is vehicular emissions [43,46,47].

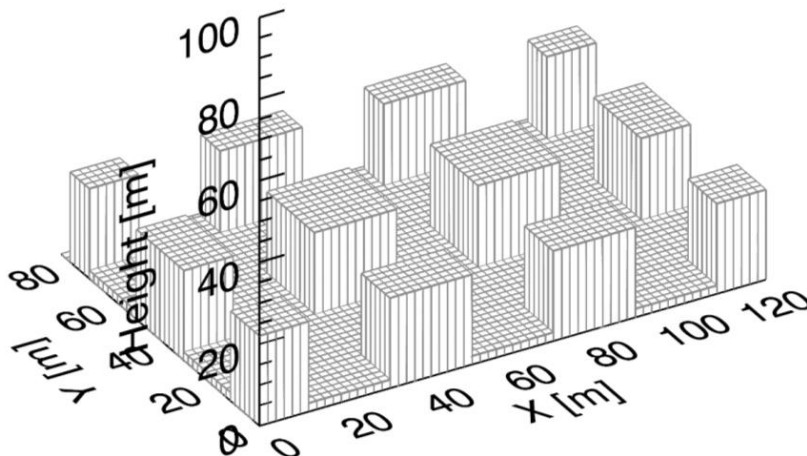

**Figure 1.** Schematic diagram of the coupled computational fluid dynamics (CFD)–chemistry simulation domain for the control run (CNTL) simulation.

**Table 1.** Emission estimates from vehicles used in the coupled CFD–chemistry model simulations for the CNTL simulation. The text discusses details about the species emission estimates.

| [g km$^{-1}$] | CO | NO$_x$ | VOC | NH$_3$ |
|---|---|---|---|---|
| PC (diesel) | 0.010 | 0.041 | 0.006 | 0.002 |
| PC (gasoline) | 0.085 | 0.004 | 0.002 | 0.023 |
| PC (gas) | 0.104 | 0.002 | 0.001 | 0.001 |
| LDV (diesel) | 0.087 | 0.063 | 0.009 | 0.002 |
| LDV (gas) | 0.163 | 0.026 | 0.002 | 0.002 |
| HDV (diesel) | 0.543 | 0.959 | 0.104 | 0.002 |
| HDV (gas) | 0.457 | 0.045 | 0.009 | 0.003 |
| Bus (diesel) | 1.850 | 0.613 | 0.150 | 0.002 |
| Bus (gas) | 1.736 | 0.346 | 0.191 | 0.000 |

**Table 2.** Ratios of vehicles following size and fuel type used in the CNTL simulation obtained from the Traffic Monitoring System.

| [%] | Passenger Cars | Light-Duty Vehicles | Heavy-Duty Vehicles | Bus |
|---|---|---|---|---|
| Diesel | 20.6 | 17.6 | 7.3 | 2.2 |
| Gasoline | 43.3 | - | - | - |
| Gas | 8.3 | 0.9 | 0.3 | 0.4 |
| Total | 72.2 | 18.5 | 7.6 | 2.6 |

For meteorological conditions, we used values for the SMR in winter that provided favorable conditions for nitrate formation [48]. To simulate diurnal changes in buoyancy and their effects on transport and chemical reaction rates in the model, we used the hourly temperature and relative humidity obtained from the Seoul station of the Korea Meteorological Administration (http://web. kma.go.kr/eng/). Table 4 shows the hourly temperature and relative humidity used in this study. The wind speed on the rooftop was assumed to be the observed seasonal mean value for SMR, which is 2.4 m s$^{-1}$. The wind direction is westerly (most frequently observed in winter) and is perpendicular to

the street canyon [49]. The ambient wind speed and direction were fixed during a one-day simulation. The following vertical profiles of the wind, turbulent kinetic energy (TKE), and TKE dissipation rate were imposed:

$$U(z) = \frac{u_* \cos\theta}{\kappa} ln\left(\frac{z}{z_0}\right) \tag{1}$$

$$V(z) = \frac{u_* \sin\theta}{\kappa} ln\left(\frac{z}{z_0}\right) \tag{2}$$

$$W(z) = 0 \tag{3}$$

$$k(z) = \frac{u_*^2}{C_\mu^{1/2}}\left(1 - \frac{z}{\delta}\right)^2 \tag{4}$$

$$\varepsilon(z) = \frac{C_\mu^{3/4} k^{3/2}}{\kappa z} \tag{5}$$

Here, $u_*$, $z_0$, and $\kappa$ represent the friction velocity, roughness length (=0.05 m), and von Karman constant (=0.4), respectively; $C_\mu$ is an empirical constant (=0.0845), and $\theta$ is the wind direction. The surface and top boundary pressures in the model were assumed to be 1013.15 hPa and 993.72 hPa, respectively.

**Table 3.** Emission rates per vehicle and ratios of speciated volatile organic compound (VOC) following the method developed by EMEP/EEA (2016).

| Tracer Name | Formula | Emission Rate [mg km$^{-1}$] | Ratio [%] |
|---|---|---|---|
| ALK4 | $\geq C_4$ alkanes | 3.60 | 24.0 |
| ISOP | $CH_2 = C(CH_3)CH = CH_2$ | - | - |
| ACET | $CH_3C(O)CH_3$ | 0.44 | 2.9 |
| MEK | $RC(O)R$ | 0.18 | 1.2 |
| ALD2 | $CH_3CHO$ | 0.98 | 6.5 |
| RCHO | $CH_3CH_2CHO$ | 1.89 | 12.6 |
| MVK | $CH_2 = CHC(=O)CH_3$ | - | - |
| MACR | $CH_2 = C(CH_3)CHO$ | - | - |
| PRPE | $\geq C_3$ alkenes | 2.58 | 17.2 |
| C3H8 | $C_3H_8$ | 0.02 | 0.1 |
| CH2O | HCHO | 1.80 | 12.0 |
| C2H6 | $C_2H_6$ | 0.05 | 0.3 |
| Unspeciated | - | - | 23.2 |

**Table 4.** Diurnal variations of the observed hourly surface temperature and relative humidity used in this model.

| Hour | 01 | 02 | 03 | 04 | 05 | 06 | 07 | 08 | 09 | 10 | 11 | 12 |
|---|---|---|---|---|---|---|---|---|---|---|---|---|
| Temperature [K] | −3.1 | −3.4 | −3.6 | −4.0 | −4.4 | −4.6 | −4.8 | −5.1 | −4.5 | −2.7 | −0.8 | 0.6 |
| Relative Humidity [%] | 37.7 | 38.3 | 37.8 | 36.8 | 37.4 | 36.9 | 37.5 | 37.6 | 32.2 | 26.4 | 21.3 | 17.9 |
| Hour | 13 | 14 | 15 | 16 | 17 | 18 | 19 | 20 | 21 | 22 | 23 | 24 |
| Temperature [K] | 1.4 | 2.2 | 2.7 | 2.6 | 1.6 | 0.3 | −0.5 | −1.2 | −1.7 | −2.1 | −2.4 | −2.8 |
| Relative Humidity [%] | 16.5 | 14.4 | 13.7 | 14.0 | 18.0 | 21.6 | 26.0 | 30.2 | 32.6 | 34.5 | 34.7 | 35.8 |

For the rooftop boundary conditions of the species, we used a reanalysis dataset from the Monitoring Atmospheric Composition and Climate (MACC) project in SMR with 6 h diurnal variations [50]. In addition, we used the boundary conditions of the species from the GEOS-Chem simulation with 6 h diurnal variations at SMR in winter for the species that were not provided in the MACC reanalysis dataset [51]. The initial conditions of the species were assumed as the concentrations of the boundary conditions in the first step. We conducted 48 h model simulations for each case: the first 24 h for the model spin-up, and the results from the last 24 h were used. The chemical and dynamical time steps were 1 min and 1 s, respectively.

Sensitivity simulations were conducted to examine the effect of emissions on nitrate aerosols in urban streets. Twelve sensitivity simulations were set by changing the vehicular emissions of $NO_x$, VOC, and $NH_3$. Each simulation was conducted with different emissions by multiplying the original emission by 0.25, 0.5, 2.0, and 4.0 for each species. We named the sensitivity simulations "species name" × "multiplying factor." For example, simulations named $NO_x$ × 0.25, $NO_x$ × 0.5, $NO_x$ × 2, and $NO_x$ × 4 indicate multiplying the vehicular $NO_x$ by 0.25, 0.5, 2.0, and 4.0, respectively, while other emissions were fixed.

### 2.3. Model Validation

The coupled CFD–chemistry model in this study was thoroughly validated for the flow and dispersion of passive tracers in street canyons by comparing the results from this model with those from a wind tunnel, an idealized numerical study, and fluid experiments [25,27,52]. Park et al. (2015) found good agreements when comparing the model with wind tunnel data and experimental data by implementing improved wall functions for the momentum and thermodynamic energy equations in the CFD model to more accurately represent the effects of solid–wall boundaries [27].

The coupled CFD model also evaluated the dispersion of reactive pollutants compared with idealized simulations and field campaigns [25]. Kim et al. (2012) applied the coupled CFD–chemistry model to simulations using the same building configuration as in Baker et al. (2004) [25,53]. Their results showed that the concentrations of $NO_x$ and $O_3$ have a pattern and magnitude consistent with the simulated concentrations by Baker et al. (2004) under steady-state $O_3$–NO–$NO_2$ photochemistry. Kim et al. (2012) reproduced reactive pollutants on Dongfeng Middle Street, Guangzhou, China, using a full tropospheric $NO_x$–$O_x$–VOC chemistry scheme and compared the results to a field campaign by Xie et al. (2003) [13]. The coupled model, with the full photochemical mechanism, also successfully captured the time variation in the observed CO concentrations for both upwind and downwind sites in the Dongfeng Street canyon. However, the coupled model overestimated $NO_x$ concentrations compared to observations by Xie et al. (2003) due to estimating excessive NO emissions from traffic volume, implying the necessity of utilizing an accurate emissions inventory [13].

The coupled CFD model has also been used to evaluate the dispersion of reactive aerosol in street canyons [28]. Kim et al. (2019) evaluated the composition of $PM_1$ in summer and winter in a street canyon by comparison with the field campaign in Elche, Spain, by Yubero et al. (2015) [28,54]. The model generally captured seasonal variations of $PM_1$ in the street canyon. We evaluated the seasonal variation in nitrate concentration by comparing our model results to those of Yubero et al. (2015) [54]. Four simulations were conducted to represent the four seasons (spring, summer, autumn, and winter) in Elche, Spain. The street was approximately 7 m wide and surrounded by buildings that were approximately 25 m in height. The domain size was 20 m × 40 m × 50 m, and the number of grid points was 42 × 82 × 52. The meteorological conditions used were the observed seasonal mean values during the campaign periods. Pollutant emissions were estimated from traffic volume obtained from the Elche Traffic Office [54]. Vehicular emissions were computed using Spain's emission rates in EMEP/EEA (2016). The detailed model configuration generally followed that of Kim et al. (2019) [28].

Figure 2 shows the observed and simulated nitrate concentrations by season. The observed nitrate concentrations were highest in winter and lowest in summer due to the thermal evaporation of nitrate aerosols, showing 1.3 and 0.3 µg m$^{-3}$, respectively. The observed nitrate concentrations in spring

and autumn were 0.4 and 0.5 $\mu g\ m^{-3}$, respectively, falling between those in winter and summer and indicating the dominant effect of temperature on nitrate aerosol. The simulated nitrate concentrations were 1.0, 0.8, 1.0, and 1.4 $\mu g\ m^{-3}$ in spring, summer, autumn, and winter, respectively, showing clear seasonal variation. The simulated concentrations reproduce the observed seasonal variation in nitrate concentration. However, the simulated nitrate concentration in summer was twice that of the observed magnitude, indicating the weaker thermal evaporation of ammonium nitrate predicted by the model. The model captured the magnitude of the observed nitrate in winter located within the standard deviation of the observed nitrate concentration. These results indicate that the model successfully calculated the nitrate aerosol in cold environments.

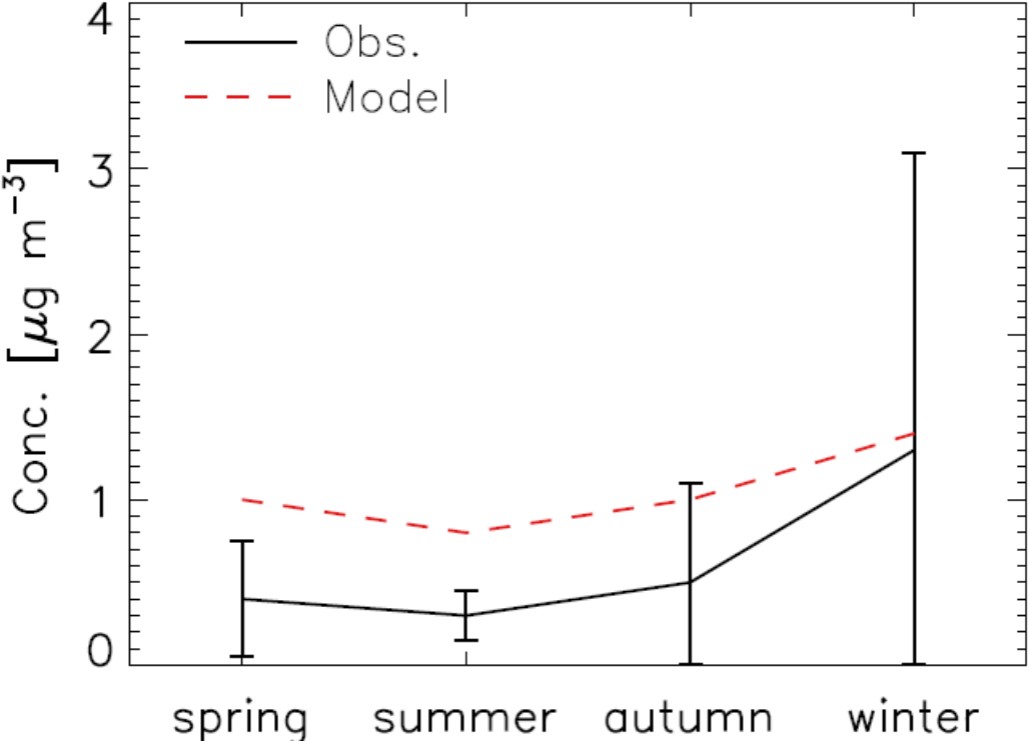

**Figure 2.** Comparisons of nitrate concentrations in the spring, summer, autumn, and winter cases between this work and the previous results of Yubero et al. (2015) (see Figure 1 in Yubero et al.). The values are averaged for the analysis period at the sample station. The black solid line is the observed concentrations and the red dashed line is the simulated concentrations. The error bars indicate the standard deviations of the observed nitrate aerosol in each season.

## 3. Results

Before investigating the production and sensitivity of nitrate aerosol, we checked the precursor gases of nitrate aerosol and oxidant concentrations that affect nitrate aerosol chemistry. Figure 3a indicates the meridionally averaged $NO_x$ concentration in the domain. The $NO_x$ concentration is highest at the street surface level, indicating that vehicular $NO_x$ emission is trapped by the canyon vortex in the street canyon. The $NO_x$ concentration reached 100 ppbv, ten times higher that outside the canyon, showing a steep gradient of $NO_x$ concentration in the street canyon. Note that concentrations in the three street canyons have slightly different values and dispersion patterns owing to different vortex patterns under non-infinite consecutive 3-D street canyons following the dispersion rates of TKE [26]. Figure 3b shows the spatial distribution of $O_3$ concentration. The $O_3$ concentration was lowest at the surface, showing a negative correlation with the $NO_x$ concentration. The $O_3$ concentrations outside and inside the street canyon were 38 and 11 ppbv, respectively, suggesting $NO_x$ titration in the street canyon. This distribution of $O_3$ and $NO_x$ fits the general dispersion pattern in the street

canyon reported in previous studies [25,27]. Figure 3c displays the $HNO_3$ concentration, which reached 3.8 ppbv in the street canyon. The $HNO_3$ concentration in the street canyon was higher than that outside the street canyon, indicating the oxidation of $HNO_3$ from vehicular $NO_x$. However, the $HNO_3$ concentration at the surface was the lowest, even though the $NO_x$ concentration was highest at the surface. The low level of $HNO_3$ at the surface was caused by low $O_3$ concentrations at the surface under VOC-limited conditions, suppressing the production of OH and $HNO_3$. Figure 3d displays the $NH_3$ concentrations; the daily averaged $NH_3$ concentration reached 2.3 ppbv, with the highest value at the surface. The dispersion pattern of $NH_3$ was similar to that of $NO_x$, indicating the high impact of vehicular emissions.

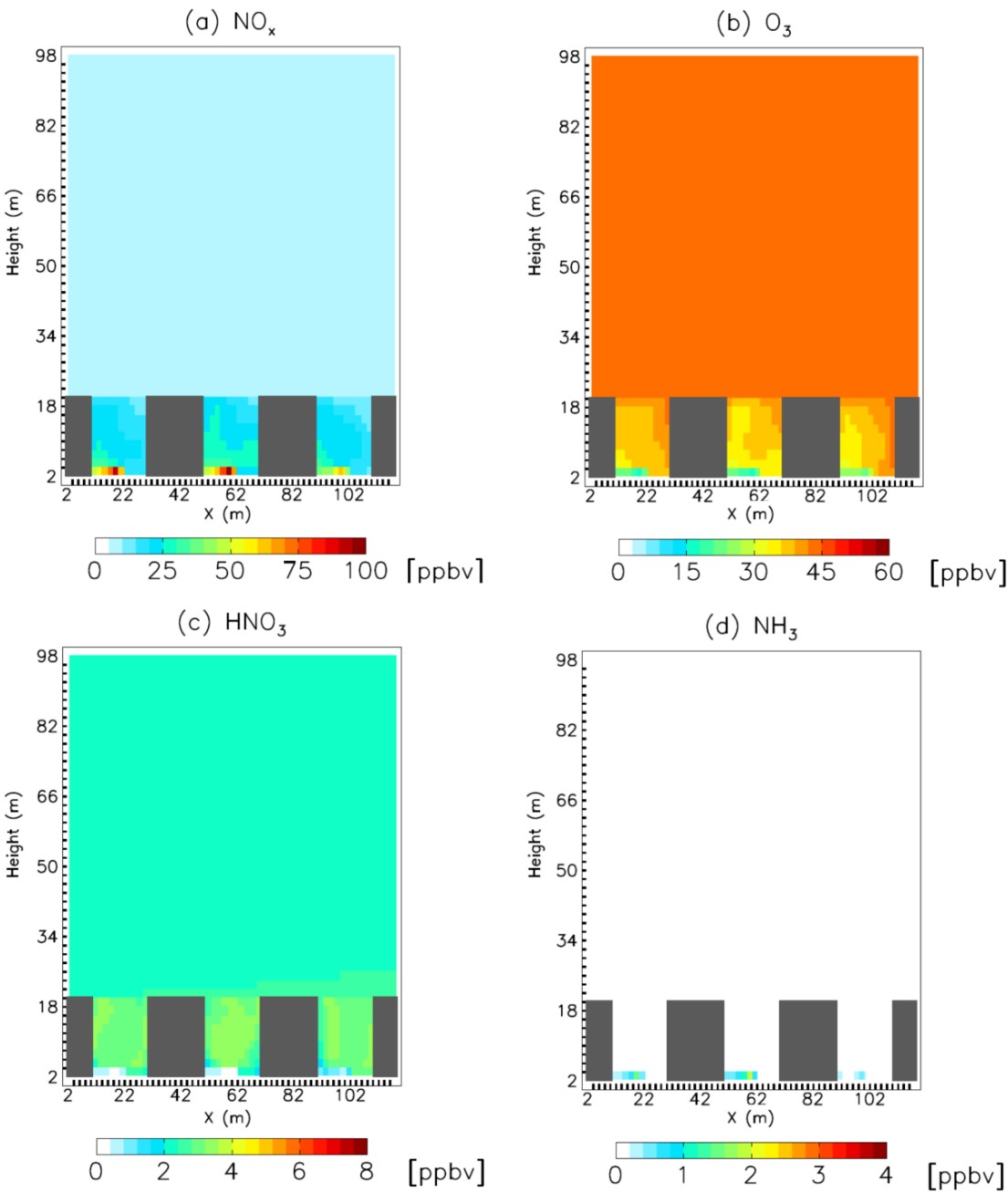

**Figure 3.** Distributions of the daily average concentrations of (**a**) $NO_x$, (**b**) $HNO_3$, (**c**) $O_3$, and (**d**) $NH_3$ (ppbv) in the CNTL simulation.

Figure 4a shows the nitrate concentrations in the street canyon, which are higher than those outside the canyon, showing values of up to 11.4 µg m$^{-3}$. The high concentration of nitrate aerosol is due to the high $HNO_3$ concentration from vehicular $NO_x$ in the street canyon. However, the spatial pattern of nitrate aerosol in the street canyon differs from that of $HNO_3$, because $NH_3$ is a precursor gas of ammonium nitrate. The nitrate aerosol was highest at the surface following vehicular $NH_3$. These high correlations between $NH_3$ and nitrate aerosol indicate that ammonium nitrate forms under $NH_3$-limited conditions in the street canyon. Figure 4b shows the ammonium concentration in the street canyon; the spatial distribution of ammonium aerosol is similar to that of nitrate, implying that most ammonium aerosols combine with nitrate aerosols in winter. The maximum concentration of ammonium aerosol was 4.7 µg m$^{-3}$, and the spatial patterns of ammonium also indicate a low concentration of ammonium sulfate in winter. Note that the vehicular emission rate of $SO_2$ was very low, implying that ammonium sulfate inside the street canyon might also be low [55]. The sum of the ammonium and nitrate concentrations was 16.1 µg m$^{-3}$, higher than the air quality guidelines set by the World Health Organization (WHO, 10 µg m$^{-3}$) and 46% of the WHO Interim Target-1 (35 µg m$^{-3}$), indicating the hazardous effect of ammonium nitrate aerosols on pedestrians [56]. Considering that nitrate chemistry is highly nonlinear, this cannot be resolved and may lead to uncertainty in regional models due to their coarser resolution.

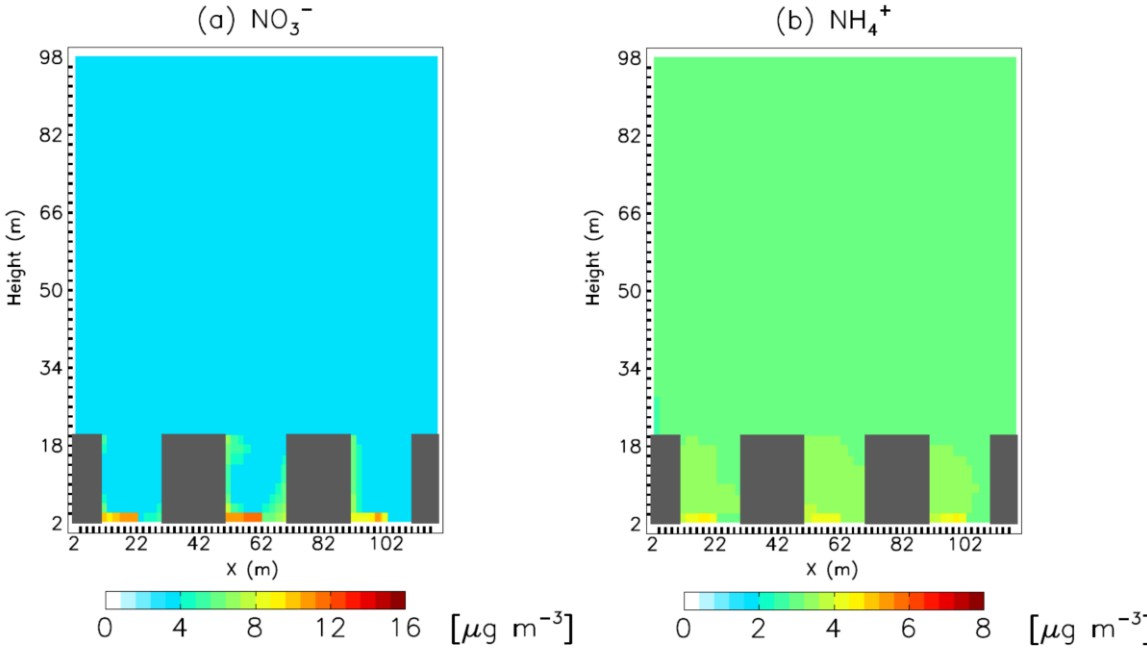

**Figure 4.** Distribution of the daily average concentrations of (**a**) nitrate and (**b**) ammonium (µg m$^{-3}$) in the CNTL simulation.

We investigated the sensitivity of nitrate aerosol production to $NO_x$ emission. Figure 5a shows the average nitrate concentration in the street canyon (i.e., below 20 m) by following vehicular $NO_x$ emission changes to investigate the sensitivity of nitrate aerosol production to the vehicular $NO_x$ emission rate. Surprisingly, the nitrate concentration did not show a clear relationship with the $NO_x$ emission rate. Nevertheless, the change in nitrate concentrations was at most 2% compared to the standard simulation, and the average nitrate concentration was highest in the CNTL and lowest in the $NO_x \times 0.25$ simulations, indicating the nonlinearity of the nitrate aerosol production to the $NO_x$ emission rate. These results contradict the conventional belief that high $NO_x$ emissions from vehicles can cause nitrate aerosol air quality problems.

Figure 5b,d display the average $HNO_3$, $O_3$, and $NO_2$ concentrations, respectively, in the street canyon according to the sensitivity simulations. The $HNO_3$ concentration in the street canyon shows

changes similar to those of nitrate aerosol, indicating that the changes in nitrate aerosol in the sensitivity simulations are closely related to the changes in $HNO_3$ (Figure 5b). The $O_3$ concentration decreases as the $NO_x$ emissions increase because of $NO_x$ titration (Figure 5d). Note that $O_3$ formation falls under a VOC-limited regime in the street canyon due to vehicular emissions. A low concentration of $O_3$ prevents the conversion of $NO_2$ to $HNO_3$ through either photochemical production during the daytime due to inhibited OH production and heterogeneous nitrate production at night. The $NO_2$ (the precursor gas of $HNO_3$), concentration in the street canyon is proportional to the $NO_x$ emissions because it affects the direct $NO_2$ emissions from vehicles and the reduction in photodissociation of $NO_2$ under low $O_3$ concentrations (Figure 5c). High $NO_2$ creates suitable conditions for $HNO_3$ formation, compensating for the effect of decreased $O_3$ on $HNO_3$. Thus, $HNO_3$ and nitrate aerosols have no clear relationship with NOx emissions and only undergo small changes. These results imply that $NO_x$ emission controls cannot improve $PM_{2.5}$ levels in urban street conditions.

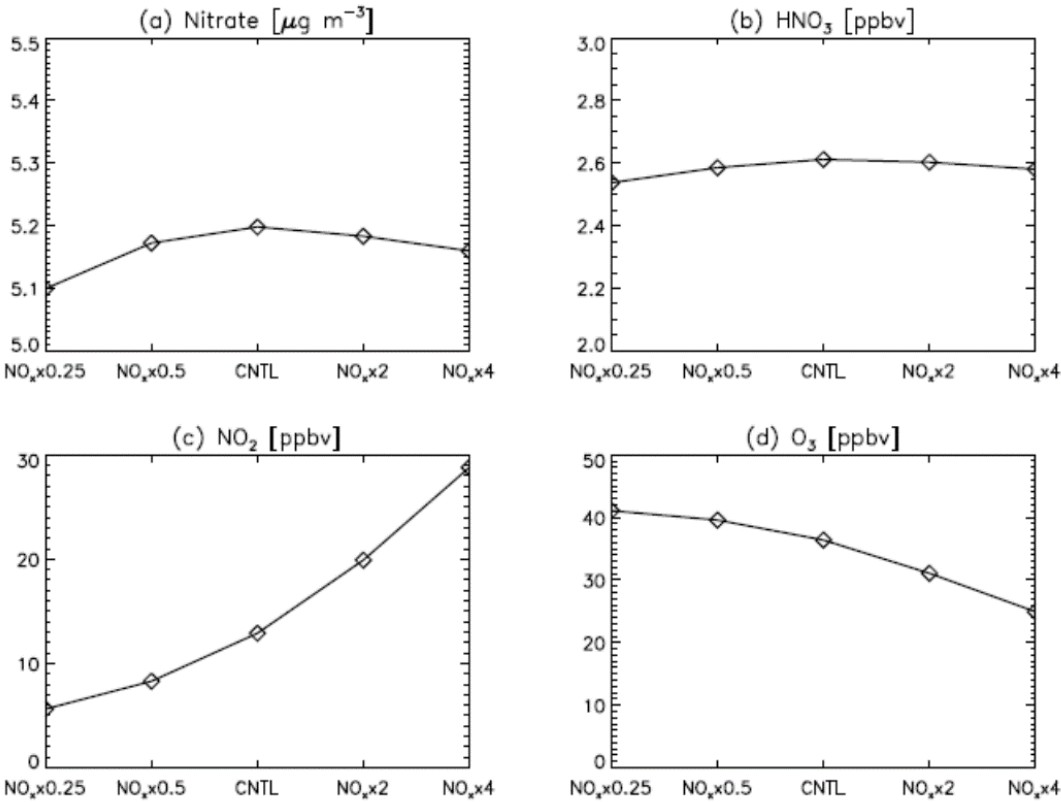

**Figure 5.** Average (**a**) nitrate, (**b**) $HNO_3$, (**c**) $NO_2$, and (**d**) $O_3$ concentrations in the street canyon (i.e., below 20 m) following a change in the emission rate of vehicular $NO_x$.

In addition, we estimated the sensitivity of nitrate aerosol production to VOC emissions. Figure 6a shows the average concentration of nitrate aerosol in the street canyon following vehicular VOC emission changes; nitrate concentrations are proportional to VOC emissions. The average nitrate concentration in the VOC × 0.25 simulation was 8% lower than that of the CNTL simulation, indicating the higher sensitivity of VOC emissions to nitrate aerosols compared with that of $NO_x$ emissions. The nitrate aerosols in VOC × 0 only showed a 12% difference with those of the CNTL simulation, which implies the large impact of the boundary condition on nitrate formation. The changes in $HNO_3$ concentration follow the changes in nitrate concentration, implying that the former is caused by the latter (Figure 6b). Reducing VOC emissions drives a decrease in both $NO_2$ and $O_3$ concentrations, creating unfavorable conditions for $HNO_3$ production, in contrast to the effect of $NO_x$ emissions (Figure 6c,d). These results are consistent with a previous box modeling study, which suggests that increases in VOC emissions induce nitrate production via $O_3$ increases under a VOC-limited regime

for $O_3$ production [57]. Considering that megacities are generally under a VOC-limited regime [58,59], VOC emission control can improve both $O_3$ control and $PM_{2.5}$ control in urban street canyons.

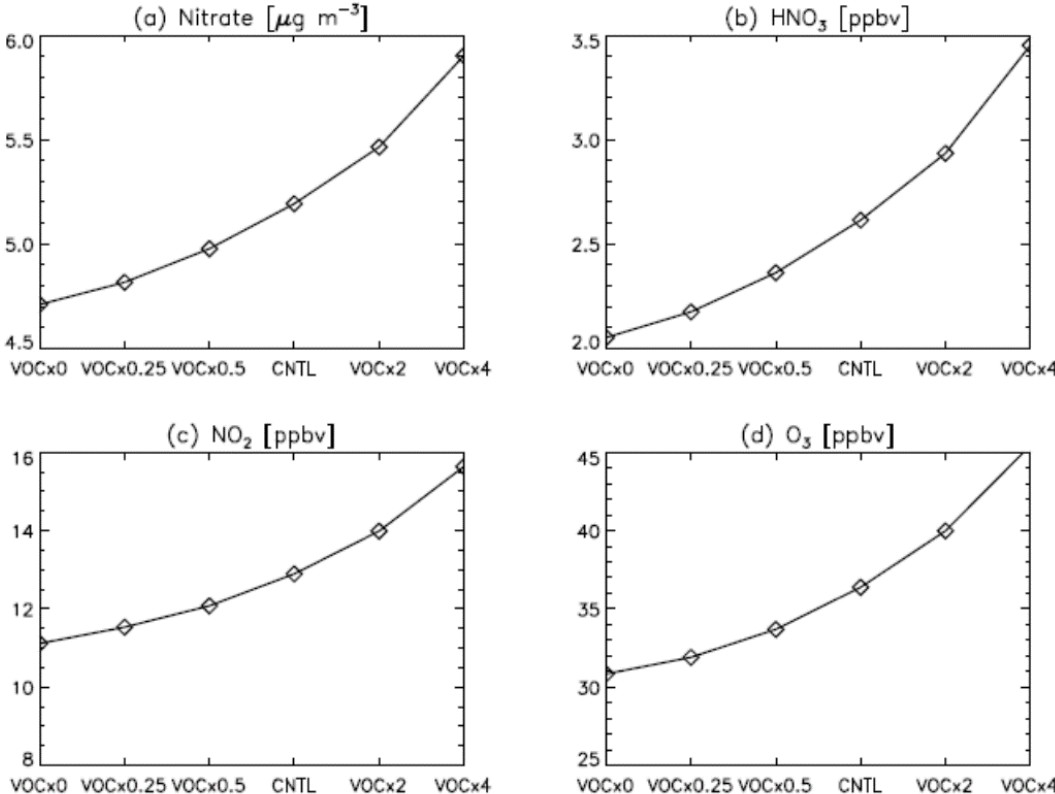

**Figure 6.** Average (**a**) nitrate, (**b**) $HNO_3$, (**c**) $NO_2$, and (**d**) $O_3$ concentrations in the street canyon (i.e., below 20 m) following changes in the emission rate of vehicular VOC.

Finally, we investigated the sensitivity of $NH_3$ emissions to nitrate production. Figure 7a shows the average concentration of nitrate aerosols in the street canyon following vehicular $NH_3$ emission changes. The concentration of nitrate aerosol was considerably influenced by the $NH_3$ emission changes; the nitrate concentration in the $NH_3 \times 4$ simulation was 42% higher than that in the standard simulation, and the nitrate concentration in $NH_3 \times 0.25$ was 85% of that in the standard simulation. These results indicate that the sensitivity of $NH_3$ emissions to nitration is much higher than that of $NO_x$ and slightly higher than that of the VOC emissions. The $HNO_3$ concentrations are inversely proportional to $NH_3$ emissions, indicating that higher $NH_3$ emissions induce a higher conversion rate of $HNO_3$ to nitrate aerosol (Figure 7b). These results suggest that the production of ammonium nitrate is reduced by the low concentration of $NH_3$ under an $NH_3$-limited regime for nitrate production. Studies based on both modeling and observed campaigns have reported that nitrate formation occurs under an $NH_3$-limited regime in East Asian megacities, including SMR [21,60]. The nitrate aerosols in the $NH_3 \times 0$ simulation were 19% lower than those with the CNTL simulation, indicating that ammonium and $NH_3$ concentrations from the boundary also have an important role in nitrate formation in the urban street canyon. These results indicate that the control of $NH_3$ emissions might be the most effective way to degrade $PM_{2.5}$ problems where vehicular emissions are dominant in winter. The regulation of vehicle emissions is mostly focused on the control of $NO_x$ emissions; considering the present findings, we should instead focus on controlling VOC and $NH_3$.

Though we used the coupled CFD–chemistry model to investigate the sensitivity of nitrate aerosols from vehicular emissions under complex geometry, our simulation still has limitations. Sea-salt aerosol significantly impacts the formation of nitrate aerosols via heterogeneous reactions when interacting with trace gases on the surface of sea-salt aerosol [61]. This process drives the efficient production

of nitrates under an $NH_3$-limited environment. This model does not account for the effect of sea-salt aerosol on nitrate aerosol production. Therefore, the simulation might underestimate the nitrate formation of heterogeneous chemistry. Moreover, we only considered the effect of vehicular emissions; $NH_3$ and VOCs emissions from heating or biogenic emissions might affect the sensitivity of nitrate formation in the street canyon. The absence of these emissions in the model domain might create uncertainty in the nitrate aerosol calculation in this simulation.

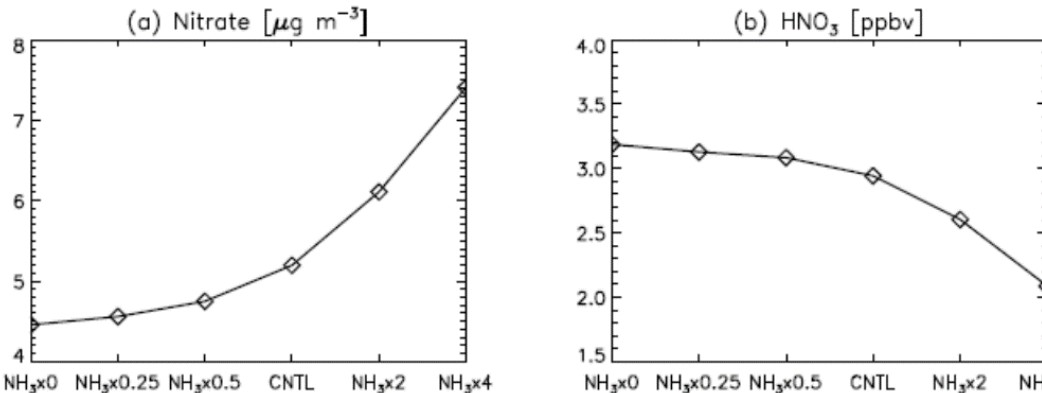

**Figure 7.** Average (**a**) nitrate and (**b**) $HNO_3$ concentrations in the street canyon (i.e., below 20 m) following a change in the emission rate of vehicular $NH_3$.

## 4. Model Sensitivity to Geometry and Speciation of VOC Emissions

We examined the sensitivity of the model to the canyon geometry by conducting sensitivity model simulations in which we changed the street canyon aspect ratios (the ratio of building height to street width) of the street canyon to 0.5 and 2.0. The conditions of the sensitivity simulations were identical to those of the CTNL simulation except that the height of the buildings, 10 m and 40 m, respectively, indicating canyon aspect ratios of 0.5 and 2.0 (Figure 8). We named the sensitivity simulations for different aspect ratios "species name" × "multiplying factor" _A "aspect ratio" (e.g., CNTL_A2.0 and $NO_x \times 2\_A0.5$). Figures 9 and 10 indicate the meridionally averaged $NO_x$, $O_3$, $HNO_3$, and nitrate aerosol concentrations in the CNTL_A0.5 and CNTL_A 2.0 simulations. The nitrate aerosol and their precursors showed similar distributions as those of the CNTL simulations despite the difference in their aspect ratios. The $NO_x$ concentration was highest at the surface and is an order of magnitude higher than that outside the canyon, indicating the trapping of vehicular emissions due to the strong canyon vortex in the street canyon (Figures 9a and 10a). The spatial distribution of $NO_x$ indicates that concentrated vehicular emissions drive the $NO_x$ titration of the $O_3$ concentration at the surface under a low VOC emissions condition in both cases (Figures 9b and 10b). The $HNO_3$ concentrations also show similar distribution to those of the CTNL simulation, indicating the suppressing of the production of OH and $HNO_3$ (Figures 9c and 10c). Figures 9d and 10d show the nitrate concentrations in the street canyon for different canyon aspect ratios. The averaged nitrate concentrations in the street canyon (<10 m and <40 m, respectively) show 5.7 and 6.2 µg m$^{-3}$ in the CNTL_A0.5 and CNTL_A 2.0 simulations, respectively, which are 9%, and 19% higher than those of the CTNL simulation. These differences are mainly due to the complex canyon vortex driving for the building geometry [27]. Despite the dispersion patterns differing according to canyon geometry, the mechanism of nitrate aerosol formation was consistent with that of CNTL, suggesting high nitrate formation from vehicular emissions in the street canyon.

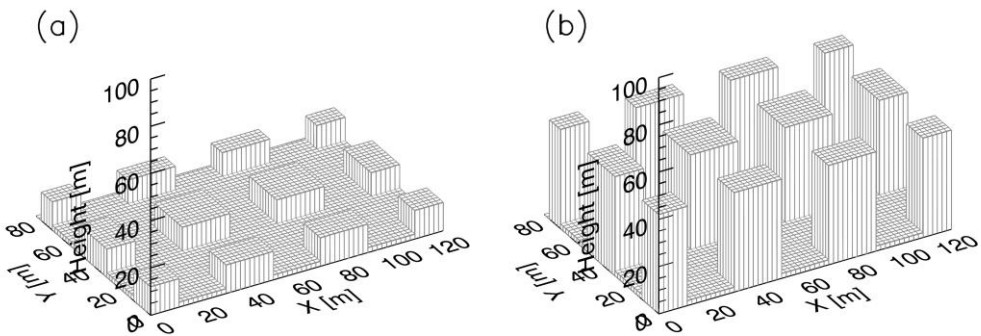

**Figure 8.** Schematic diagrams of the coupled sensitivity simulation domain for the canyon aspect ratios (**a**) 0.5 and (**b**) 2.0.

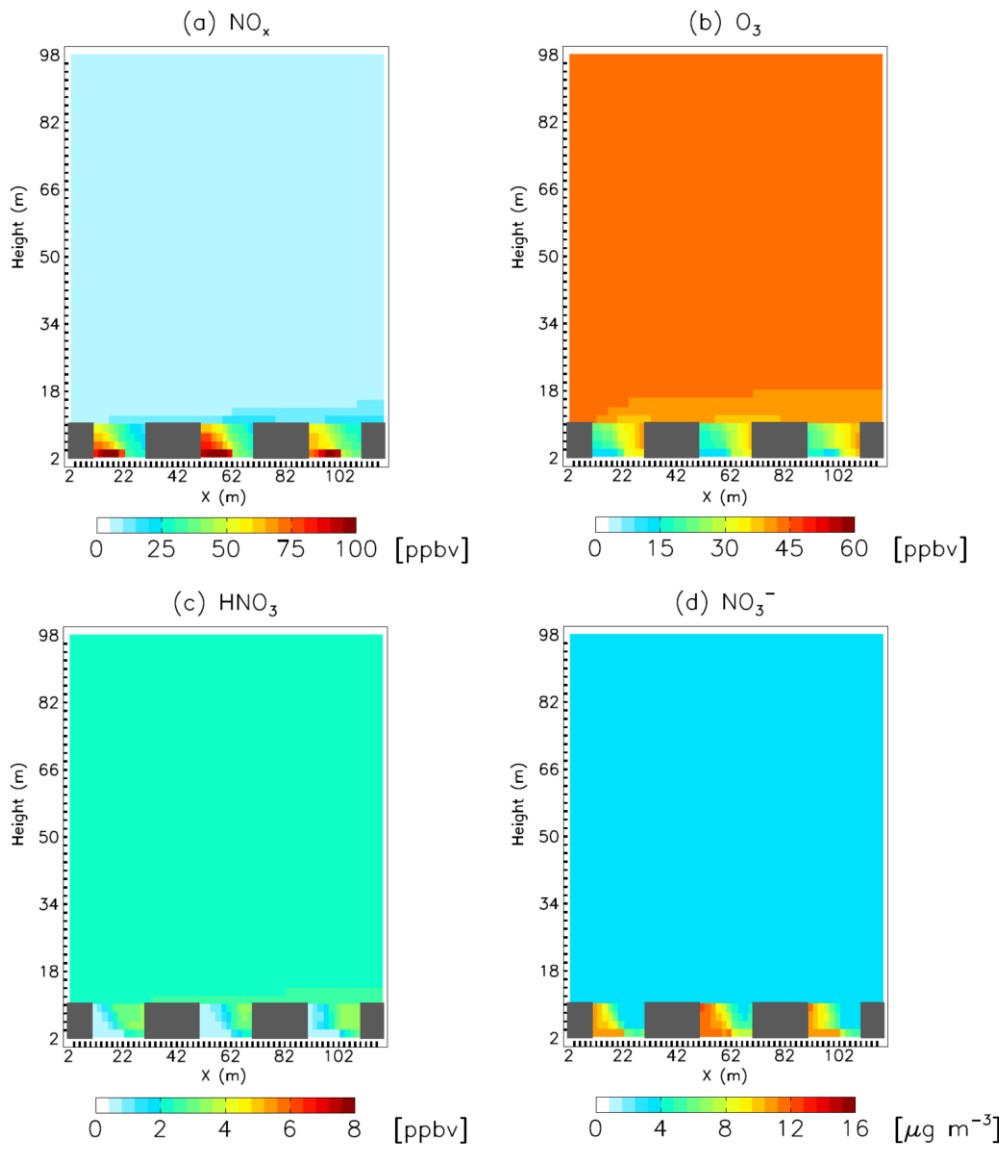

**Figure 9.** Distribution of the daily average concentrations of (**a**) $NO_x$, (**b**) $HNO_3$, (**c**) $O_3$ (ppbv), and (**d**) nitrate aerosol ($\mu g\ m^{-3}$) in the CNTL_A0.5 simulation.

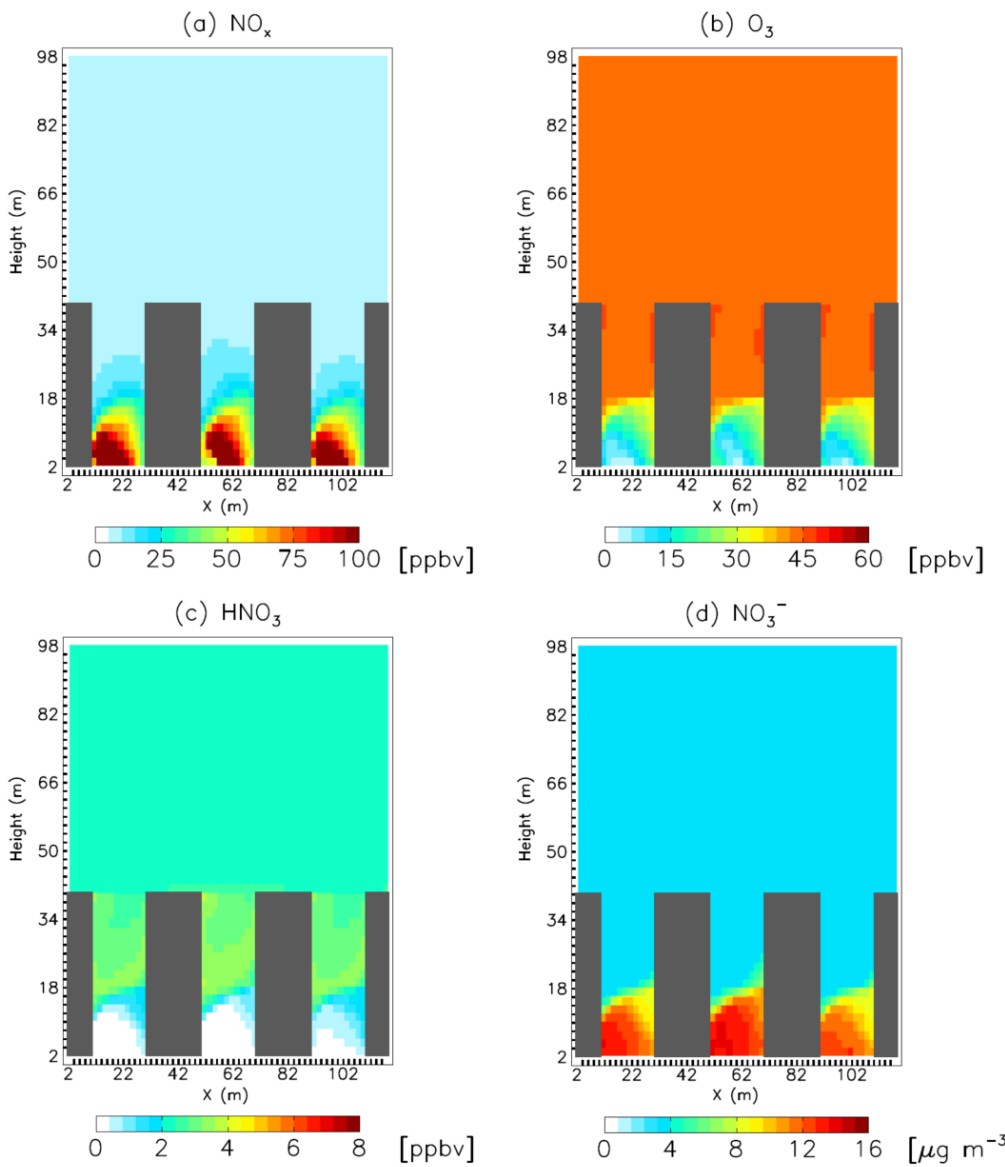

**Figure 10.** Distribution of the daily average concentrations of (**a**) $NO_x$, (**b**) $HNO_3$, (**c**) $O_3$ (ppbv), and (**d**) nitrate aerosol ($\mu g\ m^{-3}$) in the CNTL_A2.0 simulation.

We tested the $NO_x$, VOC, and $NH_3$ emission sensitivity to nitration formation for different canyon aspect ratios (0.5 and 2.0). Figures 11 and 12 show the averaged nitrate concentration in the street canyons (<10 m and <40 m, respectively) by following the vehicular $NO_x$, VOC, and $NH_3$ emission changes for the different canyon aspect ratios of 0.5 and 2.0, respectively. The sensitivity of the nitrate formation following $NO_x$, VOC, and $NH_3$ emission changes is also similar to those for the canyon aspect ratio of unity. The nitrate concentration changes show no clear relationship with the $NO_x$ emission rate in either case, which is related to the nitrate precursor changes, particularly in the daytime as we mentioned (not shown). The maximum concentrations occurred in $NO_x \times 0.5\_A0.5$ and $NO_x \times 0.5\_A2.0$, which differed slightly with the simulations for the canyon aspect ratio of unity. Nevertheless, the difference in the nitrate concentrations between the simulations was only 2%. The sensitivity of nitrate formation following VOC and $NH_3$ emissions also follows consistent results with those for the aspect ratio of unity. Enhanced (reduced) VOC emissions drive an increase (decrease) in the nitrate aerosol concentration affecting the $O_3$ concentration under a VOC limited regime for $O_3$ production. The nitrate concentration in the street canyon is proportional to the vehicular $NH_3$ emission, indicating that $NH_3$ limits the condition of nitrate formation in both cases. These results

indicate that the sensitivity of nitrate formation following emission changes in the street canyon is consistent, regardless of the building aspect ratio, due to concentrated emissions from vehicles and the canyon vortex.

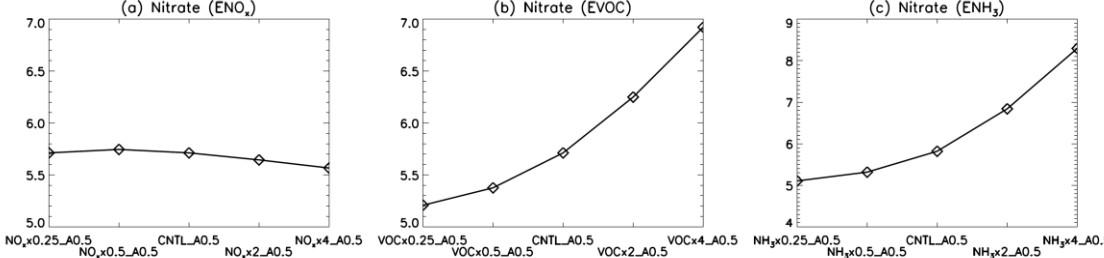

**Figure 11.** The average nitrate concentration in the street canyon following changes in the emission rate of vehicular (**a**) $NO_x$, (**b**) VOC, and (**c**) $NH_3$ for the 0.5 canyon aspect ratio; units are $\mu g\ m^{-3}$.

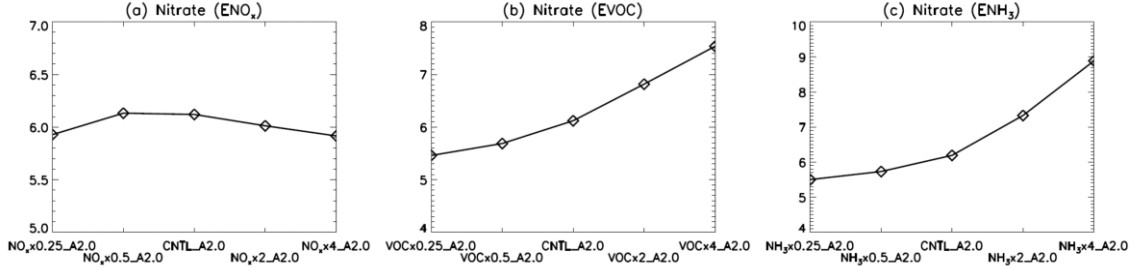

**Figure 12.** The same as Figure 11 but for the 2.0 canyon aspect ratio; units are $\mu g\ m^{-3}$.

The VOC speciation of vehicular emissions might change the model sensitivity on nitrate formation by affecting the ozone and OH production, considering all mechanisms are explained under a VOC limited regime for $O_3$ production. Therefore, we checked the sensitivity of VOC speciation of emissions on nitrate aerosol formation using the different VOC chemical speciation used by Kim et al. (2006) [62]. Table 5 summarizes the emission rates and the ratio of speciated VOC with the method of Kim et al. (2006). Figure 13 indicates the averaged nitrate concentration in the street canyon (i.e., below 20 m) by following vehicular $NO_x$, VOC, and $NH_3$ emission changes with the VOC speciation of Kim et al. (2006). Similar to other sensitivity simulations, $NO_x$ emission changes did not affect the nitrate formation (due to the conflicting effects of $NO_2$ and $O_3$) even though we changed the VOC speciation (Figure 13a). The sensitivity of VOC concentration to nitrate formation also shows a similar relationship to that with EMEP/EEA speciation. However, the sensitivity was slightly lower than the nitrate concentration in CNTL, showing only a 6% difference between VOC × 0.25 and CNTL (Figure 13b). These results show that the reduction of vehicular VOC emission is a more effective way to regulate nitrate problems in urban streets than $NO_x$ emissions under different VOC speciations.

**Table 5.** Emission rates per vehicle and **r**atios of speciated VOC obtained by following the method developed by Kim et al. (2006).

| Tracer Name | Formula | Emission Rate [mg km$^{-1}$] | Ratio [%] |
|---|---|---|---|
| ALK4 | $\geq C_4$ alkanes | 6.03 | 40.2 |
| ISOP | $CH_2 = C(CH_3)CH = CH_2$ | - | - |
| ACET | $CH_3C(O)CH_3$ | 0.15 | 1.0 |
| MEK | $RC(O)R$ | 0.0 | 0.0 |
| ALD2 | $CH_3CHO$ | 0.08 | 0.5 |
| RCHO | $CH_3CH_2CHO$ | 0.08 | 0.5 |

**Table 5.** *Cont.*

| Tracer Name | Formula | Emission Rate [mg km$^{-1}$] | Ratio [%] |
|:---:|:---:|:---:|:---:|
| MVK | $CH_2 = CHC(=O)CH_3$ | - | - |
| MACR | $CH_2 = C(CH_3)CHO$ | - | - |
| PRPE | $\geq C_3$ alkenes | 2.10 | 14.0 |
| C3H8 | $C_3H_8$ | 0.15 | 1.0 |
| CH2O | HCHO | 0.17 | 1.1 |
| C2H6 | $C_2H_6$ | 0.27 | 1.8 |
| Unspeciated | - | - | 39.9 |

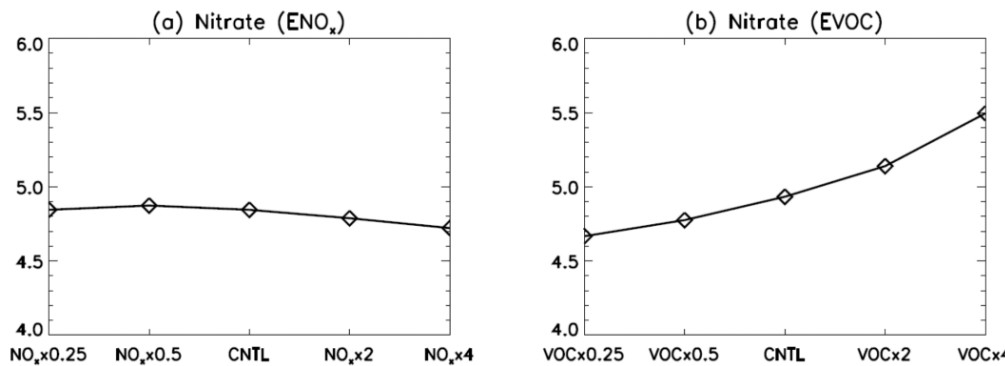

**Figure 13.** Average nitrate concentration in the street canyon following changes in the emission rate of vehicular (**a**) $NO_x$ and (**b**) VOC with the speciation method of VOC emission used by Kim et al. (2006); units are $\mu g\ m^{-3}$.

## 5. Conclusions

Nitrate contributions to $PM_{2.5}$ mass have increased in polluted urban areas, with an increasing number of severe haze events in East Asia. This study investigates the favorable conditions for the production of nitrate aerosols in urban streets using a coupled CFD–chemistry model. It was found that the nitrate concentrations in street canyons are higher than those outside the canyons due to the high $HNO_3$ concentrations from vehicular $NO_x$ in these canyons. However, the spatial pattern of nitrate aerosols in street canyons differs from that of $HNO_3$ due to $NH_3$, thus indicating that ammonium nitrate formation occurs under $NH_3$-limited conditions in street canyons.

Sensitivity simulations indicate that nitrate concentration does not show a clear relationship with the $NO_x$ emission rate, with nitrate changes of only 2% across among 16 time differences in $NO_x$ emissions. The $HNO_3$ concentration in street canyons changes in a similar manner to that of nitrate aerosols, indicating that the changes in nitrate aerosols in the sensitivity simulations are closely related to $HNO_3$ changes. An increase in the $NO_x$ emissions induces a decrease in $O_3$ and an increase in $NO_2$ under a VOC-limited regime for $O_3$ production. These changes in $O_3$ and $NO_2$ have a conflicting effect on the $HNO_3$ production in urban streets. Therefore, $HNO_3$ and nitrate aerosols have no linear relationship with $NO_x$ emissions and only undergo small changes. The sensitivity simulations were conducted by varying the vehicular VOC emissions to investigate their effect on nitrate production. The results show that nitrate concentrations are proportional to VOC emissions. Nitrate was decreased by 9% in the VOC × 0.25 simulation, indicating a relatively high sensitivity compared to that of $NO_x$. Decreased VOC emissions drive a decrease in both $NO_2$ and $O_3$ concentrations, creating unfavorable conditions for $HNO_3$ production, unlike the effect of changes in $NO_x$ emissions. The nitrate aerosol concentration is considerably influenced by $NH_3$ emissions, which show a higher sensitivity to nitrate production than do $NO_x$ and VOC emissions in urban streets. The nitrate concentration is proportional to $NH_3$ emissions with the additional production of ammonium nitrate under an $NH_3$-limited regime

for nitrate production. This research implies that, where vehicular emissions are dominant in winter, the control of vehicular VOC and NH$_3$ emissions might be a more effective way to degrade PM$_{2.5}$ problems than controlling NO$_x$.

We checked the model sensitivity by changing the model's building geometry and VOC speciation. The sensitivity of nitrate formation by following emissions changes acts in a similar direction as CNTL simulation despite changing the building geometry and speciation of VOC emissions. The sensitivity simulations revealed that our results about the sensitivity of nitrate production to emission changes are robust.

**Funding:** This work was funded by the 2017 Research Fund of Myongji University.

**Acknowledgments:** This work was supported by the 2017 Research Fund of Myongji University.

**Conflicts of Interest:** The authors declare no conflict of interest.

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
