# Peer review of "Sensitivity of Nitrate Aerosol Production to Vehicular Emissions in an Urban Street"

_atmosphere, doi:10.3390/atmos10040212_

Round 1
Reviewer 1 Report
General Comments:
Kim uses an online coupled street-level chemical transport model to investigate the source of particulate nitrate in urban street. This study develops a deeper understanding of distribution of fine particles in urban environment, which is hampered by the coarse resolution of regional models. His result reveals that particulate nitrate is most sensitive to NH3 emission from vehicles. Studies like this one with detailed focusing on street-level chemistry and air pollution are far from enough, and would be helpful for devising mitigation plans in megacities. The manuscript is well presented and in a good structure. I would like to recommend publishing this work, although there are some issues need more detailed description/discussion. Please find my specific comments following.
Specific comments:
1) N2O5 is an important source of nitrate during night, its heterogeneous hydrolysis can produce nitrate (Chen et al., 2018;Chang et al., 2016). This paper mentions a little bit about the nitrate chemistry in nighttime and heterogeneous reactions. Would you please describe in more details about how them are considered in the simulations.
2) In this paper, nitric acid neutralized by ammonia and form particulate nitrate, this is detailed discussed. Apart from this, marine air masses influence Korea frequently, sea-salt aerosol can also promote the formation of particulate nitrate in NH3-limited environment (Chen et al., 2016). Some discussion about influence of sea-salt on nitrate formation would be necessary.
3) The street-level CFD model is coupled with GEOS-Chem chemical scheme, which is designed for global simulation. A little discuss to justify the choice of GEOS-Chem scheme to represent street level chemistry would be helpful. For example, compare with CBMZ (Zaveri and Peters, 1999) which is designed for atmospheric chemistry in urban environment.
4) This paper aiming at particulate nitrate formation. More detailed description of aerosol module would be necessary. For example, sectional/modal aerosol module, how aerosol module is coupled with gaseous chemistry, how heterogeneous reactions are considered, and etc.
5) How is initial and boundary conditions considered in the street-level model? The chemical boundary conditions are provided by MACC reanalysis. What about the meteorological conditions? I am especially curious about the night period when buoyancy force from surface heating stop, could the downward transmit of turbulent kinetic energy from above rooftop be important for the street canyon? In addition, it is not clear that how the diurnal variation of T and RH applied/simulated in the model, which may have a noticeable influence on nitrate chemistry/partitioning. For example, in the night, T decreases and RH increases, which can facilitate the partitioning of nitrate from gas phase to particulate phase (Chen et al., 2018).
Minor comments:
1) line 12-13: how much change in NOx emission rate, showing nitrate changes of only 3%
2) line 16: please be specific, how can reducing vehicular VOC emissions drive the control of nitrate? What is the mechanism?
3) line 25: during haze event in which regions?
4) line 42: please use the reference as format ‘Author Year’. Please also check other places about the format issue.
5) what is the time step of the simulations.
6) line 168-169: These two paragraphs both are talking about the model validation in Elche Spain. I would suggest to combine in one paragraph.
7) line 234-236: Please consider rewording to something like:
“…is highly nonlinear, which can not be resolved and may lead to uncertainty in regional models owing to their coarser resolution.“
8) line 298-299: ‘diesel vehicles might actually emit fewer pollutants than gasoline vehicles’.
I think this statement is too arbitrary. This statement only stems from diesel vehicles might emit less NH3, which is more sensitive to nitrate formation. However, the combustion efficiency of diesel cars could be much lower than gasoline ones. I would not draw a judgment on comparison of diesel and gasoline before more comprehensive studies are conducted.
Reference:
Chang, W. L., Brown, S. S., Stutz, J., Middlebrook, A. M., Bahreini, R., Wagner, N. L., Dubé, W. P., Pollack, I. B., Ryerson, T. B., and Riemer, N.: Evaluating N2O5 heterogeneous hydrolysis parameterizations for CalNex 2010, Journal of Geophysical Research: Atmospheres, 121, 5051-5070, 10.1002/2015JD024737, 2016.
Chen, Y., Cheng, Y., Ma, N., Wolke, R., Nordmann, S., Schüttauf, S., Ran, L., Wehner, B., Birmili, W., van der Gon, H. A. C. D., Mu, Q., Barthel, S., Spindler, G., Stieger, B., Müller, K., Zheng, G. J., Pöschl, U., Su, H., and Wiedensohler, A.: Sea salt emission, transport and influence on size-segregated nitrate simulation: a case study in northwestern Europe by WRF-Chem, Atmos. Chem. Phys., 16, 12081-12097, 10.5194/acp-16-12081-2016, 2016.
Chen, Y., Wolke, R., Ran, L., Birmili, W., Spindler, G., Schröder, W., Su, H., Cheng, Y., Tegen, I., and Wiedensohler, A.: A parameterization of the heterogeneous hydrolysis of N2O5 for mass-based aerosol models: improvement of particulate nitrate prediction, Atmos. Chem. Phys., 18, 673-689, 10.5194/acp-18-673-2018, 2018.
Zaveri, R. A., and Peters, L. K.: A new lumped structure photochemical mechanism for large-scale applications, J. Geophys. Res., 104, 30387-30415, 1999.
Author Response
See attached document

Reviewer 2 Report
See attached document

Author Response
See attached document

Round 2
Reviewer 2 Report
This version of the manuscript is much improved. The author has responded to all my previous concerns, and I would now recommend publication.
This manuscript is a resubmission of an earlier submission. The following is a list of the peer review reports and author responses from that submission.